# Search for anomalous quartic gauge couplings in the process $\mu^+\mu^- \to \bar{\nu}\nu\gamma\gamma$ with a nested local outlier factor

**Ke-Xin Chen[1,2], Yu-Chen Guo[1,2] and Ji-Chong Yang[1,2]⋆**

**1** Department of Physics, Liaoning Normal University, No. 850 Huanghe Road, Dalian 116029, China
**2** Center for Theoretical and Experimental High Energy Physics, Liaoning Normal University, No. 850 Huanghe Road, Dalian 116029, China

⋆ yangjichong@lnnu.edu.cn

## Abstract

In recent years, with the increasing luminosities of colliders, handling the growing amount of data has become a major challenge for future new physics (NP) phenomenological research. To improve efficiency, machine learning algorithms have been introduced into the field of high-energy physics. As a machine learning algorithm, the local outlier factor (LOF), and the nested LOF (NLOF) are potential tools for NP phenomenological studies. In this work, the possibility of searching for the signals of anomalous quartic gauge couplings (aQGCs) at muon colliders using the NLOF is investigated. Taking the process $\mu^+\mu^- \to \nu\bar{\nu}\gamma\gamma$ as an example, the signals of dimension-8 aQGCs are studied, expected coefficient constraints are presented. The NLOF algorithm are shown to outperform the k-means based anomaly detection methods, and a tradition counterpart.

## Contents

## 1 Introduction

18 As the Large Hadron Collider (LHC) experiment transitions into the post-Higgs discovery phase,
19 physicists have embarked on the quest for new physics (NP) beyond the Standard Model (SM) [1,
20 2], which is widely believed to be exist at higher energy scales. The pursuit of NP has emerged as
21 a leading frontier in high energy physics (HEP) research. HEP investigations frequently entail the
22 analysis of extensive datasets stemming from particle collisions or other experimental endeavors.
23 Given the fact that, in the foreseeable next decade or so, upgrades of colliders will focus primar-
24 ily on luminosity rather than energy, the efficiency of data analyzing becomes a more and more
25 important topic.

26 Machine learning (ML) promises to substantially speeding up data processing and analysis,
27 thereby serving as a pivotal tool in advancing the detection of NP signals. To efficiently analyze
28 data within this context, previous studies have applied anomaly detection (AD) ML algorithms
29 in the field of HEP for the purpose of searching for NP signals. [3–25]. In AD algorithms, one
30 notable method is the local outlier factor (LOF) [26]. As a density-based AD algorithm, it can
31 be expected to effectively screen signals of NP when combined with the nested anomaly detec-
32 tion algorithm [27], even when interference terms play a significant role. Compared to the nested
33 isolation forest method used in Refs. [27], an additional motivation for our investigation of LOF's
34 effectiveness lies in that, the core computation in LOF primarily involves calculating point-to-point
35 distances. Even when extended with nesting, as in the nested LOF (NLOF) algorithm proposed
36 in this study, the computational backbone remains anchored in distance calculations. This grants
37 both LOF and NLOF inherent flexibility, i.e., we can strategically define various kernel func-
38 tions, precompute inter-point distances, and subsequently input them within (N)LOF frameworks.
39 Notably, with the recent surge of quantum ML applications in NP searches, quantum comput-
40 ing as a high-throughput data processing paradigm, enables ultra-efficient distance computation
41 through quantum kernels. This naturally facilitates quantum-enhanced extensions, quantum kernel
42 (N)LOF in the future. As an unsupervised ML algorithm, the LOF can automatically discover the
43 anomalies, which is even useful if there was no NP signals, because it can be expected that the
44 signal of rare processes or the possible artifacts of the colliders can emerge as anomalous signals.
45 When NLOF is introduced, although the algorithm needs a reference dataset from the SM, it does
46 not need information about the NP models, i.e., it can search for NP without knowing what NP
47 model it is searching for.

48 As a validation, we consider the Standard Model Effective Field Theory (SMEFT) [28–33].
49 The prominence of SMEFT stems precisely from its applicability in high-luminosity regimes
50 where collision energies remain below the threshold required to directly excite NP degrees of
51 freedom, making it inherently aligned with this study's focus. Concurrently, the muon collider is
52 considered as the experimental scenario [34–42]. As a lepton collider, it offers high luminosity
53 and relatively clean QCD backgrounds. It is worth noting that, as a geometrically well-defined
54 algorithm, the (N)LOF algorithms are fundamentally agnostic to the specific NP models under in-
55 vestigation, and should remain universally applicable across arbitrary collider configurations and
56 theoretical frameworks.

57 The muon collider is recognized as an effective gauge boson collider, particularly suited for
58 probing vector boson scattering (VBS) processes. Among these, the $WW \to \gamma\gamma$ channel stands out
59 as a prominent VBS candidate due to its distinct advantages, including absence of forward-moving
60 charged leptons, and no additional electroweak (EW) vertices. As a consequence, this study fo-

cuses on the process $\mu^+\mu^- \to \nu\bar{\nu}\gamma\gamma$. As a VBS process, the process $\mu^+\mu^- \to \nu\bar{\nu}\gamma\gamma$ is suitable to study the SMEFT operators contributing to anomalous quartic gauge couplings (aQGCs) [43–47]. High dimensional operators generating aQGCs independent of anomalous triple gauge couplings emerge starting at dimension-8 in the SMEFT. Therefore the signals of dimension-8 aQGCs operators in the $\mu^+\mu^- \to \nu\bar{\nu}\gamma\gamma$ is adopted in this work, which thus serves as a timely complement to the growing interest in dimension-8 operator analyses [48–58], and directly aligning with the current focus in NP phenomenology that prioritizes precision EW measurements and high-dimensional operator disentanglement at future colliders.

It is worth clarifying that if the goal were solely to identify anomalous events, one would not need to know the NP model. However, if no trace of NP were found, the purpose of NP phenomenology becomes constraining the parameters of NP. To achieve this, introducing an NP model becomes necessary. In our phenomenological study, not only the aQGCs are introduced, but also an event selection strategy designed to maximize the signal is adopted, which incorporates supervised learning. To investigate the impact of developing event selection criteria using only background events, the scenarios are also considered where the number of remaining background events is 1%, 5%, and 10%.

The remainder of the paper is organized as follows. In section 2, a brief introduction to aQGCs and the $\mu^+\mu^- \to \nu\bar{\nu}\gamma\gamma$ process is given. The event selection strategy of (N)LOF is discussed in section 3. Section 4 presents numerical results for the expected coefficient constraints. Section 5 is a summary of the conclusions.

## 2    aQGCs and the process $\mu^+\mu^- \to \nu\bar{\nu}\gamma\gamma$ at the muon colliders

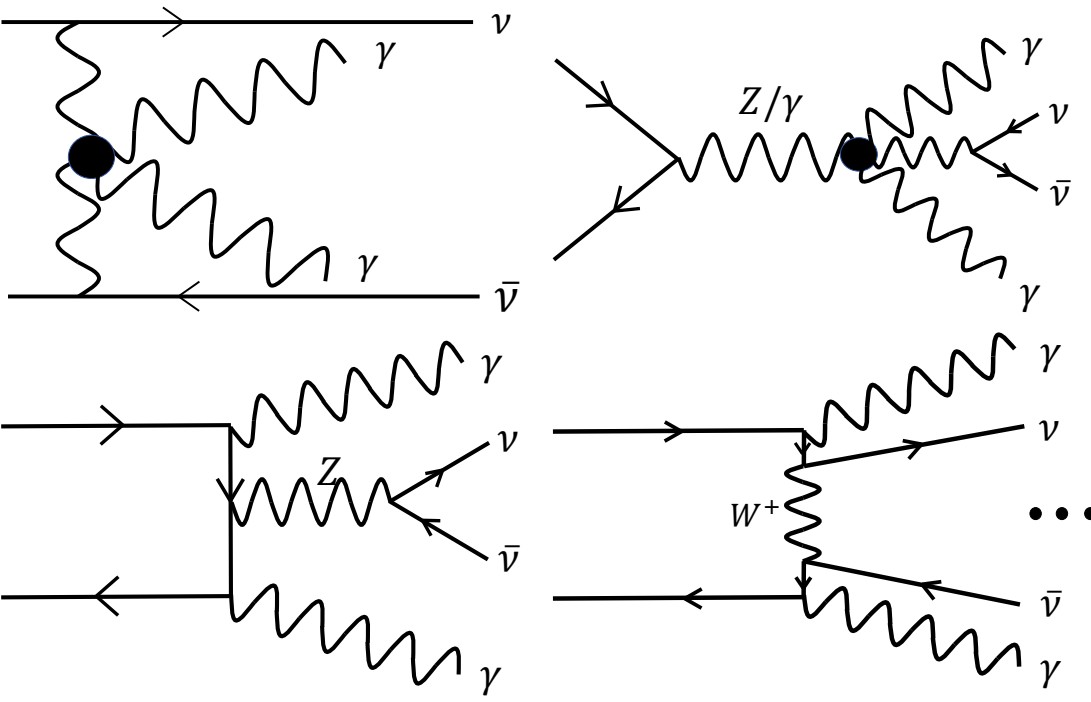

Figure 1: Typical Feynman diagrams for signal events (the upper panels) and background events (the lower panels).

A key concern in NP phenomenological studies is the sensitivity of processes at (future) col-

liders to NP models. To form a cross-reference with other studies [59–74], we use the most commonly used set of dimension-8 aQGCs operators [75, 76]. Only the operators mixed transverse and longitudinal operators $O_{M_i}$ and the transverse operators $O_{T_i}$ in this operator set are involved in the process $\mu^+\mu^- \to \nu\bar{\nu}\gamma\gamma$, the Lagrangian is,

$$\mathcal{L}_{\text{aQGC}} = \sum_i \frac{f_{M_i}}{\Lambda^4} O_{M_i} + \sum_j \frac{f_{T_j}}{\Lambda^4} O_{T_j}, \tag{1}$$

where $f_{M_i}$ and $f_{T_j}$ are dimensionless Wilson coefficients, and $\Lambda$ is the NP energy scale. The operators $O_{M_{0,1,2,3,4,5,7}}$ and $O_{T_{0,1,2,5,6,7}}$ can contribute to the process of $\mu^+\mu^- \to \nu\bar{\nu}\gamma\gamma$ at muon colliders,

$$
\begin{aligned}
O_{M_0} &= \text{Tr}\left[\widehat{W}_{\mu\nu}\widehat{W}^{\mu\nu}\right] \times \left[\left(D^\beta\Phi\right)^\dagger D^\beta\Phi\right], \\
O_{M_1} &= \text{Tr}\left[\widehat{W}_{\mu\nu}\widehat{W}^{\nu\beta}\right] \times \left[\left(D^\beta\Phi\right)^\dagger D^\mu\Phi\right], \\
O_{M_2} &= \left[B_{\mu\nu}B^{\mu\nu}\right] \times \left[(D_\beta\Phi)^\dagger D^\beta\Phi\right], \\
O_{M_3} &= \left[B_{\mu\nu}B^{\nu\beta}\right] \times \left[(D_\beta\Phi)^\dagger D^\mu\Phi\right], \\
O_{M_4} &= \left[(D_\mu\Phi)^\dagger \widehat{W}_{\alpha\nu} D^\mu\Phi\right] \times B^{\beta\nu}, \\
O_{M_5} &= \left[\left(D_\mu\Phi\right)^\dagger \widehat{W}_{\beta\nu} D_\nu\Phi\right] \times B^{\beta\mu} + h.c., \\
O_{M_7} &= \left(D_\mu\Phi\right)^\dagger \widehat{W}_{\beta\nu}\widehat{W}_{\beta\mu} D_\nu\Phi, \\
O_{T_0} &= \text{Tr}\left[\widehat{W}_{\mu\nu}\widehat{W}^{\mu\nu}\right] \times \text{Tr}\left[\widehat{W}_{\alpha\beta}\widehat{W}^{\alpha\beta}\right], \\
O_{T_1} &= \text{Tr}\left[\widehat{W}_{\alpha\nu}\widehat{W}^{\mu\beta}\right] \times \text{Tr}\left[\widehat{W}_{\mu\beta}\widehat{W}^{\alpha\nu}\right], \\
O_{T_2} &= \text{Tr}\left[\widehat{W}_{\alpha\mu}\widehat{W}^{\mu\beta}\right] \times \text{Tr}\left[\widehat{W}_{\beta\nu}\widehat{W}^{\nu\alpha}\right], \\
O_{T_5} &= \text{Tr}\left[\widehat{W}_{\mu\nu}\widehat{W}^{\mu\nu}\right] \times B_{\alpha\beta}B^{\alpha\beta}, \\
O_{T_6} &= \text{Tr}\left[\widehat{W}_{\alpha\nu}\widehat{W}^{\mu\beta}\right] \times B_{\mu\beta}B^{\alpha\nu}, \\
O_{T_7} &= \text{Tr}\left[\widehat{W}_{\alpha\mu}\widehat{W}^{\mu\beta}\right] \times B_{\beta\nu}B^{\nu\alpha},
\end{aligned}
\tag{2}
$$

where $\widehat{W} \equiv \vec{\sigma}\cdot\vec{W}/2$ with $\sigma$ being the Pauli matrices and $\vec{W} = \{W^1, W^2, W^3\}$, $B_\mu$ and $W_\mu^i$ stand for the gauge fields of $U(1)_Y$ and $SU(2)_I$, $B_{\mu\nu}$ and $W_{\mu\nu}$ are field strength tensors, and $D_\mu\Phi$ is the covariant derivative. The typical Feynman diagrams are shown in Fig. 1. The absence of forward-moving charged leptons in the final state, as well as the fact that the final state of the VBS subprocess are two photons which avoids the introduction of additional EW vertices, make it well suited for exploring the aQGCs. At the muon colliders, the VBS processes are associated with very energetic muons or neutrinos in the forward region with respect to the beam. One important background could be the beam induced background, especially those with the final states containing soft/collinear photons/muons which can escape from the detectors and mimic the neutrinos. It is challenging to generate events with soft/collinear final states using Monte Carlo (MC) methods, as they often lead to infrared divergences, making such backgrounds difficult to study, especially at the current stage when the detector has not yet been constructed. In this work, we neglect the contribution from these backgrounds.

As an effective field theory, the SMEFT is only valid under the NP energy scale. The high center-of-mass (c.m.) energy achievable at muon colliders offers an excellent opportunity to detect potential NP signals. However, it is necessary to verify the validity of the SMEFT framework.

| $\sqrt{s}$ | 3 TeV | 10 TeV |
|---|---|---|
| $f_{M_0}/\Lambda^4$ $(\mathrm{TeV}^{-4})$ | 8.2 | $6.6 \times 10^{-2}$ |
| $f_{M_1}/\Lambda^4$ $(\mathrm{TeV}^{-4})$ | 32.7 | $2.6 \times 10^{-1}$ |
| $f_{M_2}/\Lambda^4$ $(\mathrm{TeV}^{-4})$ | 1.2 | $1.0 \times 10^{-2}$ |
| $f_{M_3}/\Lambda^4$ $(\mathrm{TeV}^{-4})$ | 4.9 | $3.9 \times 10^{-2}$ |
| $f_{M_4}/\Lambda^4$ $(\mathrm{TeV}^{-4})$ | 4.5 | $3.6 \times 10^{-2}$ |
| $f_{M_5}/\Lambda^4$ $(\mathrm{TeV}^{-4})$ | 9.0 | $7.3 \times 10^{-2}$ |
| $f_{M_7}/\Lambda^4$ $(\mathrm{TeV}^{-4})$ | 65.4 | $5.3 \times 10^{-1}$ |
| $f_{T_0}/\Lambda^4$ $(\mathrm{TeV}^{-4})$ | 1.9 | $1.5 \times 10^{-2}$ |
| $f_{T_1}/\Lambda^4$ $(\mathrm{TeV}^{-4})$ | 5.7 | $4.6 \times 10^{-2}$ |
| $f_{T_2}/\Lambda^4$ $(\mathrm{TeV}^{-4})$ | 7.6 | $6.1 \times 10^{-2}$ |
| $f_{T_5}/\Lambda^4$ $(\mathrm{TeV}^{-4})$ | 0.57 | $4.6 \times 10^{-3}$ |
| $f_{T_6}/\Lambda^4$ $(\mathrm{TeV}^{-4})$ | 1.7 | $1.4 \times 10^{-2}$ |
| $f_{T_7}/\Lambda^4$ $(\mathrm{TeV}^{-4})$ | 2.3 | $1.8 \times 10^{-2}$ |

Table 1: The tightest partial wave unitarity bounds at $\sqrt{s} = 3$ TeV and 10 TeV.

Partial wave unitarity has been extensively employed in previous studies as a criterion for assessing the validity of the SMEFT [77–85]. Partial-wave unitarity bounds are process-specific. When using an EFT to study a process at a given energy scale, if the Wilson coefficients are sufficiently large such that the amplitude violates unitarity, it indicates that the EFT fails to provide a consistent description of the process under those coefficients and at that energy. For the subprocess $WW \to \gamma\gamma$, in the c.m. frame with $z$-axis along the flight direction of $W^-$ in the initial state, the helicity amplitude can be expanded using the Wigner d-function as [77],

$$\mathcal{M}(W^-_{\lambda_1} W^+_{\lambda_2} \to \gamma_{\lambda_3}\gamma_{\lambda_4}) = 8\pi \sum_J (2J+1)\sqrt{1+\delta_{\lambda_3\lambda_4}} e^{i(\lambda-\lambda')\phi} d^J_{\lambda\lambda'}(\theta) T^J, \tag{3}$$

where $\lambda_{1,2} = \pm 1, 0$ and $\lambda_{3,4} = \pm 1$ correspond to the helicities of the vector bosons, $\theta$ and $\phi$ are zenith and azimuth angles of $\gamma_{\lambda_3}$, $\lambda = \lambda_1 - \lambda_2$, $\lambda' = \lambda_3 - \lambda_4$, and $T_J$ is the coefficient of the expansion. The partial wave unitarity bound is $|T^J| \leq 2$ [80]. The results of the helicity amplitudes as well as the $|T^J|$ are calculated in Ref. [24], the tightest bounds for each operators are obtained. For the subprocess $WW \to \gamma\gamma$, the results of the partial wave unitarity bounds for one operator at a time are [24],

$$\left|\frac{f_{M_0}}{\Lambda^4}\right| \leq \frac{128\sqrt{2}\pi M_W^2}{\hat{s}^2 e^2 v^2}, \qquad \left|\frac{f_{M_1}}{\Lambda^4}\right| \leq \frac{512\sqrt{2}\pi M_W^2}{\hat{s}^2 e^2 v^2},$$

$$\left|\frac{f_{M_2}}{\Lambda^4}\right| \leq \frac{64\sqrt{2}\pi M_W^2 s_W^2}{\hat{s}^2 e^2 v^2 c_W^2}, \qquad \left|\frac{f_{M_3}}{\Lambda^4}\right| \leq \frac{256\sqrt{2}\pi M_W^2 s_W^2}{\hat{s}^2 e^2 v^2 c_W^2},$$

$$\left|\frac{f_{M_4}}{\Lambda^4}\right| \leq \frac{128\sqrt{2}\pi M_W^2 s_W}{\hat{s}^2 e^2 v^2 c_W}, \qquad \left|\frac{f_{M_5}}{\Lambda^4}\right| \leq \frac{256\sqrt{2}\pi M_W^2 s_W}{\hat{s}^2 e^2 v^2 c_W},$$

$$\left|\frac{f_{M_7}}{\Lambda^4}\right| \leq \frac{1024\sqrt{2}\pi M_W^2}{\hat{s}^2 e^2 v^2}, \tag{4}$$

$$\left|\frac{f_{T_0}}{\Lambda^4}\right| \leq \frac{8\sqrt{2}\pi}{\hat{s}^2 s_W^2}, \qquad \left|\frac{f_{T_1}}{\Lambda^4}\right| \leq \frac{24\sqrt{2}\pi}{\hat{s}^2 s_W^2},$$

$$\left|\frac{f_{T_2}}{\Lambda^4}\right| \leq \frac{32\sqrt{2}\pi}{\hat{s}^2 s_W^2}, \qquad \left|\frac{f_{T_5}}{\Lambda^4}\right| \leq \frac{8\sqrt{2}\pi}{\hat{s}^2 c_W^2},$$

$$\left|\frac{f_{T_6}}{\Lambda^4}\right| \leq \frac{24\sqrt{2}\pi}{\hat{s}^2 c_W^2}, \qquad \left|\frac{f_{T_7}}{\Lambda^4}\right| \leq \frac{32\sqrt{2}\pi}{\hat{s}^2 c_W^2},$$

118 where $\sqrt{\hat{s}}$ is the c.m. energy of the subprocess and must be less equal to $\sqrt{s}$. Therefore, the
119 strongest constraints can be obtained by using $\sqrt{s}$ instead of $\sqrt{\hat{s}}$ in Eq. (4), and the numerical
120 results at $\sqrt{s} = 3$ and 10 TeV are listed in Table 1.

# 3   The event selection strategy of NLOF

122 The searching for NP signals at a high luminosity collider involves sifting through vast datasets to
123 identify a small number of anomalous events. The LOF is an algorithm designed to find a small
124 number of anomalous events based on density. Therefore, it is reasonable to expect that the LOF
125 is suitable for the search of NP. Apart from this, since LOF algorithm is based on density, it also
126 suits the nested AD (NAD) event selection strategy proposed in Ref. [27] which is useful when
127 the interference between the SM and NP is important.
128      The core in the calculation of (N)LOF is to compute the distances. In the case of phenomeno-
129 logical studies in HEP, there are different ways to define the distance between two events (denoted
130 as $d(A, B)$ where $A$ and $B$ are the points in the feature space to which the events $A$ and $B$ is
131 mapped) [27, 86]. In this work, we use the Euclidean distance, i.e. $d(A, B)$ is the Euclidean dis-
132 tance between two points representing the two events in the feature space. However, the definition
133 of $d(A, B)$ can be regarded as a kernel function, and different definitions may yield optimizations
134 of the performance, and it can also be replaced by quantum kernels in future researches.

## 3.1   A brief introduction of LOF

136 LOF introduces a concept 'local reachability density' (LRD) which can be viewed as a measure-
137 ment of density in its neighborhood. And a point is likely an outlier if its density is significantly
138 smaller than the average density of its neighbors. To calculate LRD, LOF introduces a concept
139 'reachability distance' (LD) which can be viewed as an analogous of distance. Then $LRD = 1/\overline{LD}$,
140 where $\overline{LD}$ is the average of LD between the point and its neighbors. That is, if the point is far away
141 from its neighbors, it is considered to be in a sparse region (low density region). The detailed pro-
142 cedure can be spited as follows,

143     1. For a point $A$, compute its distance to its k-th nearest neighbor (denoted as $kd(A)$). Identify
144       its k-nearest neighbors (denoted as $kNN(A)$).

145     2. For each neighbor of $A$ (denoted as $B$), calculate the LD between them as $LD(A, B) = \max\{d(A, B), kd(B)\}$.

146     3. For $A$ and its neighbors, compute their LRD as, $LRD(A) = k / \sum_{B \in kNN(A)} LD(A, B)$.

147     4. Calculate the LOF score (denoted as $a$) as $a(A) = \left( \sum_{B \in kNN(A)} LRD(B)/k \right) / LRD(A)$.

148 After the anomaly score $a$ is obtained, one can use $a > a_{th}$ as a criterion to select NP signal events,
149 where $a_{th}$ is a tunable threshold. In LOF, there is another tunable parameter $k$, both the choice of
150 $k$ and $a_{th}$ which will be discussed in the next section.

## 3.2   Using NLOF to search for aQGCs

152 In the relatively low energy region, the difference between the kinematic characteristics of the
153 signaling event and the SM becomes less significant. In this scenario, finding NP signals is no
154 longer a problem of AD. The NLOF selection event strategy is introduced to address this problem.
155 Since the anomaly score computed by the LOF algorithm can be regarded as a measure of the
156 density of events in the feature space, it can be inferred that the anomaly score can also be used
157 to measure changes in density, which is the idea of NAD. In NAD, one construct a reference of
158 anomaly scores based on the SM background events, and use the changes of anomaly scores to

159  select events which are obtained by comparing with the reference set. The NLOF event selection
160  strategy can be summarized as follows,

1. Using the dataset obtained from MC simulations of the SM as the training dataset (denoted
   as $S_r$), the LOF applied to obtain the anomaly score for each event, denoted as $a_r$.

2. For the dataset to be investigated (it can be from the MC simulation or from the experiments,
   denoted as $S_i$), the LOF is again applied to obtain the anomaly score for each event, denoted
   $a_i$.

3. For each event in $S_i$, find the nearest neighbor event in $S_r$ and calculate the change in the
   anomaly score as $\Delta a = a_i - a_r$.

168  After $\Delta a$ is obtained, one can use $|\Delta a| > \Delta a_{th}$ as a criterion to select NP signal events.

## 4  Numerical result

### 4.1  Data preparation

| $\sqrt{s}$ | 3 TeV |
|---|---|
| $f_{M_3}/\Lambda^4$ (TeV$^{-4}$) | $[-2.7, 2.7]$ [74] |
| $f_{M_4}/\Lambda^4$ (TeV$^{-4}$) | $[-3.7, 3.6]$ [74] |
| $f_{M_5}/\Lambda^4$ (TeV$^{-4}$) | $[-8.3, 8.3]$ [87] |
| $f_{T_0}/\Lambda^4$ (TeV$^{-4}$) | $[-0.12, 0.11]$ [71] |
| $f_{T_1}/\Lambda^4$ (TeV$^{-4}$) | $[-0.12, 0.13]$ [71] |
| $f_{T_2}/\Lambda^4$ (TeV$^{-4}$) | $[-0.28, 0.28]$ [71] |
| $f_{T_5}/\Lambda^4$ (TeV$^{-4}$) | $[-0.31, 0.33]$ [74] |
| $f_{T_6}/\Lambda^4$ (TeV$^{-4}$) | $[-0.25, 0.27]$ [74] |
| $f_{T_7}/\Lambda^4$ (TeV$^{-4}$) | $[-0.67, 0.73]$ [74] |

Table 2: The range of coefficients in the case where the partial wave unitarity bounds are
looser than the constraints at the LHC.

171  The events are generated by scanning in the coefficient space within unitarity bounds and the
172  constraints obtained at 95% C.L. at the LHC. The constraints at the LHC for $O_{M_{0,1,7}}$ operators are
173  tight (the constraints in Ref. [71] are one order of magnitude than the unitarity bounds in Table 1),
174  and the signals at the $\sqrt{s} = 3$ (TeV) can be hardly observed if we use the range of the coefficients at
175  the LHC, and therefore these operators are not studied in this work. When the unitarity bounds are
176  tighter, we use the range in Table 1, otherwise, we use the coefficient ranges listed in Table 2. The
177  simulation is carried out with the help of the MC simulation toolkits `MadGraph5@NLO` [88–90].
178  A fast detector simulation is applied by using the `Delphes` [91] with the default muon collider
179  card. The signal and background events are prepared using `MLAnalysis` [92], and the anomaly
180  scores are calculated using the LOF algorithm in `scikit-learn` [93]. Since the unitarity bounds
181  are strong for the $O_{M_i}$ operators at $\sqrt{s} = 10$ TeV, and there are few signal events when scanning
182  the coefficients within the limit of the unitarity bounds, or a larger number of background evens is
183  needed, for simplicity, at $\sqrt{s} = 10$ TeV only $O_{T_i}$ operators are considered.
184  For the purpose of investigating the signal events, it is required that the final state contains at
185  least two photons. The axes of the feature space are chosen to be five observables including $E_{\gamma_1}$,
186  $p_{\gamma_1}^T$, $E_{\gamma_2}$, $p_{\gamma_2}^T$ and $m_{\gamma\gamma}$, where $E_{\gamma_{1,2}}$ are the energies of the hardest and the second hardest photons,
187  $p_{\gamma_{1,2}}^T$ are the transverse momenta of them, and $m_{\gamma\gamma}$ is the invariant mass of them. By neglecting the

effect of the detector simulation, except for the azimuth angles the information of the momenta of photons as well as the missing momentum can be reproduced by these five observables. As a ML algorithm, the NLOF is expected to automatically adapt to different feature spaces, which exhibit no fundamental performance differences but demonstrate variations in discriminability. All components of momenta are also tested as the feature space, producing comparable results with reduced computational efficiency. These five variables are selected for optimized computational cost. In order to reduce the 12-dimensional feature space to 5 dimensions, we analyzed the final state and selected five observables as the feature space. Such dimensionality reduction can, in fact, be achieved in a process-agnostic manner using ML algorithms, automatically and without the need to analyze the underlying physics. For instance, autoencoders or principle component analysis (PCA) can be employed for data dimensionality reduction. For complex processes with a large number of final-state particles, where manual analysis for dimensionality reduction becomes particularly intricate, this approach is especially meaningful. After dimensionality reduction, the z-score standardization [94] is applied to these observables, namely, $\hat{p}_i = (p_i - \bar{p}_i)/\varepsilon_i$, where $p_i$ denotes the i-th observable of an event, $\bar{p}_i$, $\varepsilon_i$ are the average and standard deviation of $p_i$ over the SM dataset. Then the feature space is a five-dimensional space composed of these five observables after the z-score standardization. An event (the j-th event) can be mapped to a point in this five-dimensional feature space as $\{\hat{p}_i^j\}$.

## 4.2 Compare the LOF with NLOF

Although LOF is inherently designed for exploring anomalous signals such as the ones from the NP, we find that it struggles to confine the expected coefficients within the partial wave unitarity bounds. Taking the case where LOF demonstrates relatively good performance as an example, we present a comparative analysis between LOF and NLOF for $O_{M_2}$ at $\sqrt{s} = 3$ TeV. We tried all three cases with $k = 500$, 1000 and 2000, and the NLOF rendering is best at $k = 2000$. And it is the LOF rendering that is best at $k = 2000$, so in the use of NLOF and LOF two algorithms are taken $k = 2000$ respectively. Results with different choices of $k$ and thresholds will be discussed in the next subsection.

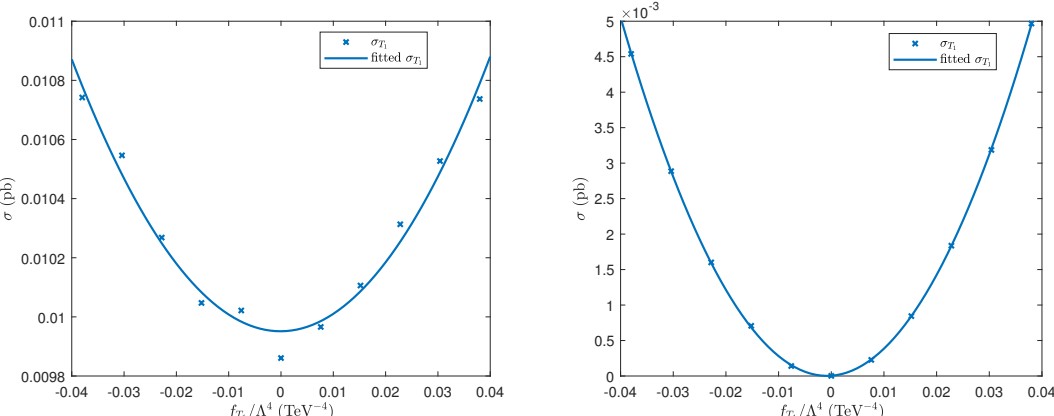

Figure 2: Fitted cross sections after cut using the LOF (the left panel) and the NLOF (the right panel) algorithms at $\sqrt{s} = 10$ TeV for $O_{T_1}$.

After scanning the coefficient space, the anomaly scores $a$ are calculated, then a cut $a > a_{th} = 1.5$ (optimized for the signal significance) is applied. With interference between the SM and NP considered, the cross section after cut is,

$$\sigma(f) = \sigma_{SM} + f\sigma_{int} + f^2\sigma_{NP} \tag{5}$$

where $f$ is the operator coefficient, $\sigma_{SM}$, $\sigma_{int}$ and $\sigma_{NP}$ are parameters to be fitted representing the contribution from the SM, the interference, and the NP alone, respectively.

The fitting of the cross section after cut in the case of LOF is shown in the left panel of Fig. 2. For NLOF, after selecting the events with $\Delta a > \Delta a_{th} = 0.08$ (optimized for the signal significance), the cross section is also fitted according to Eq. (5), the result is shown in the right panel of Fig. 2. It can be seen that for the case of NLOF, the signal is more significant.

The expected coefficient constraints can be estimated using the signal significance defined as [95, 96],

$$\mathcal{S}_{stat} = \sqrt{2\left[(N_{\mathrm{bg}} + N_s)\ln(1 + N_s/N_{\mathrm{bg}}) - N_s\right]}, \tag{6}$$

where $N_{bg}$ is the event numbers of the background and $N_s$ is the event numbers of the signal background, $N_{bg} = \sigma_{\mathrm{SM}}L$ and $N_s = \left(f\sigma_{int} + f^2\sigma_{NP}\right)L$, where $f$ is the constraint to be solved, $\sigma_{\mathrm{SM}}$, $\sigma_{int}$, and $\sigma_{NP}$ are the parameters fitted according to Eq. (5), and $L$ is the luminosity. The expected constraints at $2\sigma$, $3\sigma$ and $5\sigma$ can be obtained by solving the equations $\mathcal{S}_{stat} = 2, 3$ and 5.

At $\sqrt{s} = 3$ TeV the designed luminosity is $L = 1$ ab [97, 98]. The expected coefficient constraints at $\mathcal{S}_{stat} = 2$ are $[-1.12, 1.19]$ (TeV$^{-4}$) in the case of LOF, and $[-0.267, 0.284]$ (TeV$^{-4}$) in the case of NLOF. It can be seen that, the NLOF can outperform the LOF by one order of magnitude.

## 4.3 Expected constraints on the coefficients

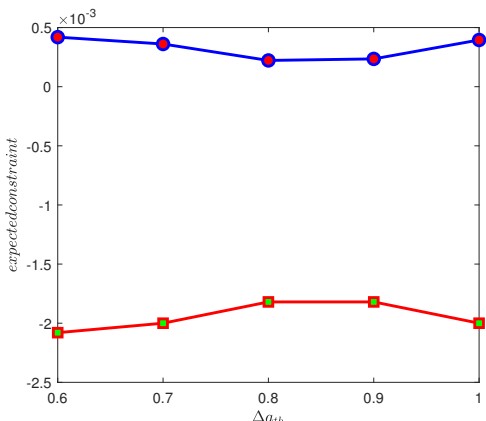

Figure 3: Expected constraint on the coefficient $f_{T_1}/\Lambda^4$ as a function of $\Delta a_{th}$ for the NLOF at $\sqrt{s} = 10$ TeV.

The expected constraints in the cases of $O_M$ operators at $\sqrt{s} = 3$ TeV and $O_T$ operators at both $\sqrt{s} = 3$ TeV and $\sqrt{s} = 10$ TeV are studied. The $\Delta a_{th}$ are chosen as 0.08 at $\sqrt{s} = 3$ TeV and 0.8 at $\sqrt{s} = 10$ TeV, respectively. It has been introduced that, the thresholds of the anomaly scores to select the events are optimized for signal significance. As an example, the expected constraints on $O_{T_1}$ at $\sqrt{s} = 10$ TeV as a function of $\Delta a_{th}$ is shown in Fig. 3. The fittings of the cross sections after cuts are shown in Figs. 4 and 5. The cross sections for the cases of $O_{M_3}$ and $O_{M_4}$ are close to each other due to the accident that, at leading order of $M_Z^2/s$, the NP contributions for the two cases are $\sigma_{O_{M_3}}/\sigma_{O_{M_4}} = 17c_W^2/(60s_W^2) \approx 1$ [24]. The expected constraints on the coefficients obtained using signal significance are shown in Tables 3 and 4.

In addition to different thresholds, the selection of different $k$-values also influences the results. In Table 5, we present a comparison of results for $k = 500, 1000$, and 2000. It can be observed that larger $k$-values yield tighter coefficient constraints. Even larger $k$-values are not considered

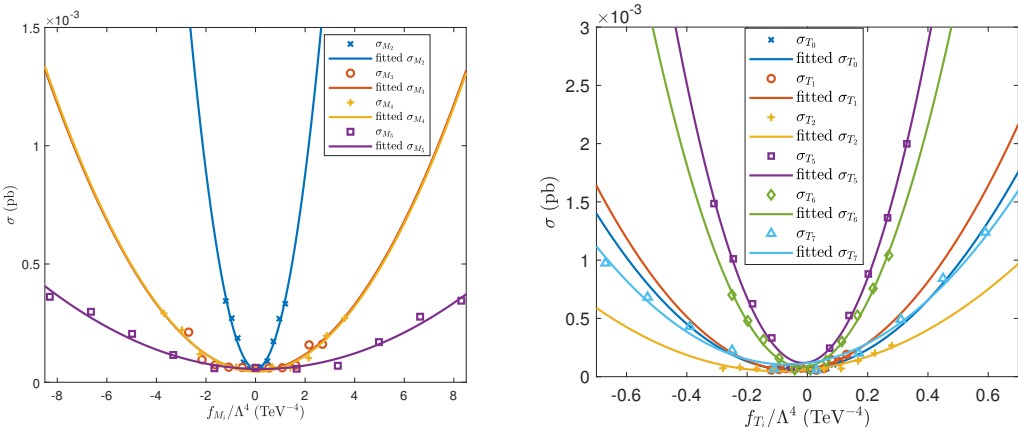

Figure 4: Fitted cross sections after cut at $\sqrt{s} = 3$ TeV for $O_{M_{2,3,4}}$ (the left panel) and $O_{T_{0,1,2,5,6,7}}$ (the right panel).

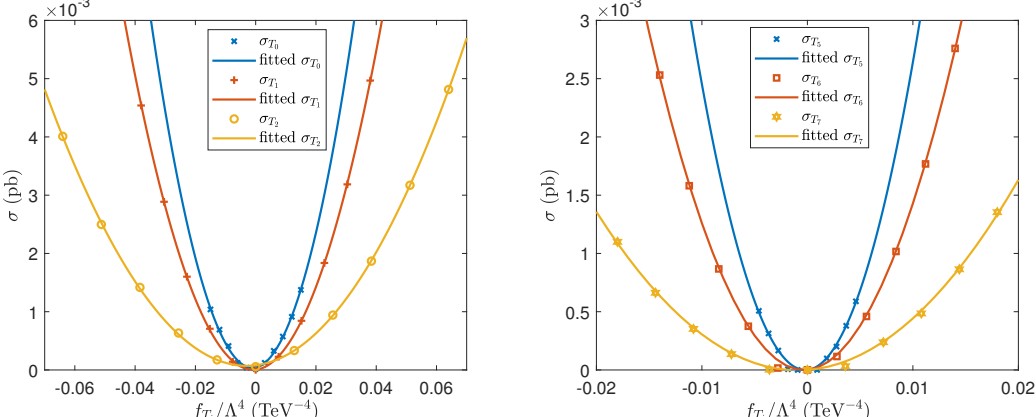

Figure 5: Fitted cross sections after cut at $\sqrt{s} = 3$ TeV for $O_{T_{0,1,2}}$ (the left panel) and $O_{T_{5,6,7}}$ (the right panel).

| | $S_{stat}$ | 3 TeV 1 ab$^{-1}$ (TeV$^{-4}$) | | $S_{stat}$ | 3 TeV 1 ab$^{-1}$ (TeV$^{-4}$) |
|---|---|---|---|---|---|
| $\frac{f_{M_2}}{\Lambda^4}$ | 2 | $[-0.267, 0.284]$ | $\frac{f_{M_3}}{\Lambda^4}$ | 2 | $[-0.898, 0.912]$ |
| | 3 | $[-0.333, 0.349]$ | | 3 | $[-1.11, 1.13]$ |
| | 5 | $[-0.440, 0.457]$ | | 5 | $[-1.47, 1.48]$ |
| $\frac{f_{M_4}}{\Lambda^4}$ | 2 | $[-0.845, 0.940]$ | $\frac{f_{M_5}}{\Lambda^4}$ | 2 | $[-1.64, 2.07]$ |
| | 3 | $[-1.06, 1.15]$ | | 3 | $[-2.07, 2.51]$ |
| | 5 | $[-1.41, 1.50]$ | | 5 | $[-2.79, 3.22]$ |
| $\frac{f_{T_0}}{\Lambda^4}$ | 2 | $[-0.124, 0.0415]$ | $\frac{f_{T_1}}{\Lambda^4}$ | 2 | $[-0.134, 0.0333]$ |
| | 3 | $[-0.139, 0.0565]$ | | 3 | $[-0.147, 0.0464]$ |
| | 5 | $[-0.165, 0.0825]$ | | 5 | $[-0.170, 0.0693]$ |
| $\frac{f_{T_2}}{\Lambda^4}$ | 2 | $[-0.231, 0.0471]$ | $\frac{f_{T_5}}{\Lambda^4}$ | 2 | $[-0.0533, 0.0265]$ |
| | 3 | $[-0.251, 0.0665]$ | | 3 | $[-0.0617, 0.0348]$ |
| | 5 | $[-0.285, 0.101]$ | | 5 | $[-0.0756, 0.0487]$ |
| $\frac{f_{T_6}}{\Lambda^4}$ | 2 | $[-0.0620, 0.0261]$ | $\frac{f_{T_7}}{\Lambda^4}$ | 2 | $[-0.183, 0.0479]$ |
| | 3 | $[-0.0708, 0.0348]$ | | 3 | $[-0.201, 0.0663]$ |
| | 5 | $[-0.0855, 0.0496]$ | | 5 | $[-0.233, 0.0981]$ |

Table 3: Projected sensitivity the coefficients of the $O_{M_{2,3,4}}$ and $O_{T_{0,1,2,5,6,7}}$ operators at $\sqrt{s} = 3$ TeV.

| | $S_{stat}$ | 10 TeV 10 ab$^{-1}$ ($10^{-4}$TeV$^{-4}$) | | $S_{stat}$ | 10 TeV 10 ab$^{-1}$ ($10^{-4}$TeV$^{-4}$) |
|---|---|---|---|---|---|
| $\frac{f_{T_0}}{\Lambda^4}$ | 2 | $[-13.2, 0.510]$ | $\frac{f_{T_1}}{\Lambda^4}$ | 2 | $[-18.2, 2.22]$ |
| | 3 | $[-13.5, 0.820]$ | | 3 | $[-19.2, 3.23]$ |
| | 5 | $[-14.2, 1.51]$ | | 5 | $[-21.1, 5.12]$ |
| $\frac{f_{T_2}}{\Lambda^4}$ | 2 | $[-67.5, 8.03]$ | $\frac{f_{T_5}}{\Lambda^4}$ | 2 | $[-3.98, 0.265]$ |
| | 3 | $[-71.0, 11.5]$ | | 3 | $[-4.13, 0.420]$ |
| | 5 | $[-77.3, 17.8]$ | | 5 | $[-4.46, 0.753]$ |
| $\frac{f_{T_6}}{\Lambda^4}$ | 2 | $[-6.96, 0.658]$ | $\frac{f_{T_7}}{\Lambda^4}$ | 2 | $[-18.9, 0.599]$ |
| | 3 | $[-7.29, 0.988]$ | | 3 | $[-19.3, 0.941]$ |
| | 5 | $[-7.95, 1.64]$ | | 5 | $[-20.0, 1.69]$ |

Table 4: Same as Table 3 but for $\sqrt{s} = 10$ TeV.

| k | | $S_{stat}$ | 10 TeV 10 ab$^{-1}$ ($10^{-4}$TeV$^{-4}$) |
|---|---|---|---|
| 500 | $\frac{f_{T_1}}{\Lambda^4}$ | 2 | $[-20.1, 5.17]$ |
| | | 3 | $[-22.0, 7.13]$ |
| | | 5 | $[-25.4, 10.5]$ |
| 1000 | $\frac{f_{T_1}}{\Lambda^4}$ | 2 | $[-20.0, 4.49]$ |
| | | 3 | $[-21.8, 6.26]$ |
| | | 5 | $[-24.9, 9.35]$ |
| 2000 | $\frac{f_{T_1}}{\Lambda^4}$ | 2 | $[-18.2, 2.22]$ |
| | | 3 | $[-19.2, 3.23]$ |
| | | 5 | $[-21.1, 5.12]$ |

Table 5: When k = 500, 1000 and 2000, under the condition of $\sqrt{s} = 10$ TeV,Projected sensitivity the coefficients of the $O_{T_1}$ operators.

due to computational resource limitations. However, based on the comparison in Table 5, it can be inferred that the sensitivity of the coefficient constraints to *k* is moderate.

## 4.4 Compare of NLOF with other methods

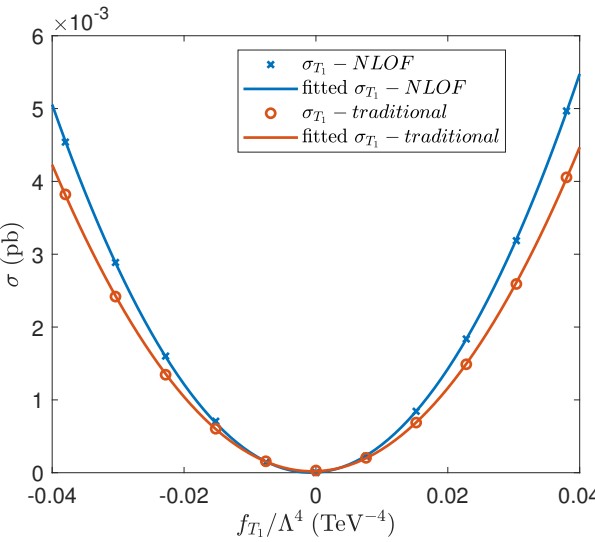

Figure 6: Comparison of the cross section after cut between the case of a traditional event selection strategy and the NLOF for $O_{T_1}$ at $\sqrt{s} = 10$ TeV.

In Ref. [24] the signals of aQGCs in the process $\mu^+\mu^- \to \gamma\gamma\nu\bar{\nu}$ at $\sqrt{s} = 10$ TeV was also considered, but with the interference terms ignored, with a kmeans AD (KMAD) algorithm and a quantum kernel KMAD (QKMAD) algorithm. To compare our method with QKMAD, we use the operator $O_{T_1}$ at $\sqrt{s} = 10$ TeV and $S_{stat} = 2$ as an example. A traditional event selection strategy is also include in the comparison, which is,

$$p_{\gamma_1}^T > 2.2 \text{ TeV}, \quad p_{\gamma_2}^T < 0.8 \text{ TeV}, \quad m_{\gamma\gamma} > 1 \text{ TeV}. \tag{7}$$

The fitting of the traditional event selection strategy is compared with NLOF in Fig. 6, it can be

shown that the NLOF can preserve more signal events while suppressing the background events to a similar amplitude as the traditional event selection strategy.

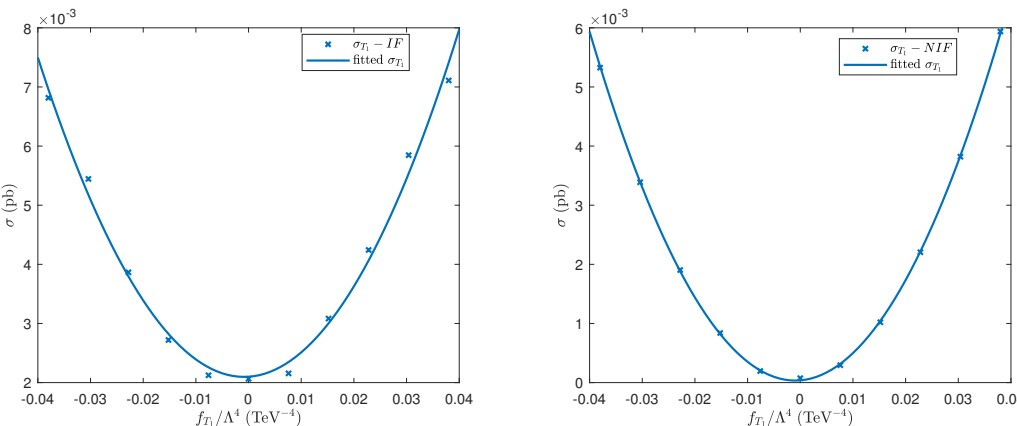

Figure 7: Same as Fig. 2 but for the IF (the left panel) and the NIF (the right panel).

Isolation forest (IF) was used in Ref. [99] to study the signals of aQGCs in VBS processes at the LHC. Nested IF (NIF) was employed in Ref. [27] to investigate the signals of neutral triple gauge couplings (nTGCs) in diboson processes at the CEPC. In this work, we also utilize IF and NIF to study the $O_{T_1}$ operator in the process $\mu^+\mu^- \to \gamma\gamma\nu\bar{\nu}$ at $\sqrt{s} = 10$ TeV. The same dataset is used, and the forest consists of 100 trees. For IF, we select the events with $\Delta a > \Delta a_{th} = 0.6$ (optimized for the signal significance), where $a$ is the anomaly score in the sense of IF. For NIF, we select the events with $\Delta a > \Delta a_{th} = 0.05$ (optimized for the signal significance), where $\Delta a$ is the change of anomaly score in the sense of NIF. The cross sections after cuts, along with the fittings are shown in Fig. 7. The expected coefficient constraint calculated by the traditional event selection strategy is $[-1.73 \times 10^{-3}, 6.15 \times 10^{-4}]$ (TeV$^{-4}$), by the NLOF algorithm is $[-1.82 \times 10^{-3}, 2.22 \times 10^{-4}]$ (TeV$^{-4}$), by the IF algorithm is $[-3.84 \times 10^{-3}, 2.15 \times 10^{-3}]$ (TeV$^{-4}$), by the NIF algorithm is $[-2.34 \times 10^{-3}, 4.38 \times 10^{-4}]$ (TeV$^{-4}$), by KMAD is $[-1.66 \times 10^{-3}, 1.66 \times 10^{-3}]$ (TeV$^{-4}$), and by QKMAD is $[-1.65 \times 10^{-3}, 1.65 \times 10^{-3}]$ (TeV$^{-4}$). It can be seen that the expected coefficient constraint of the NLOF algorithm is the tightest among all methods. The comparison of the above mentioned methods are shown in the left panel of Fig. 8.

To make the comparison more transparent, and to study the impact of developing event selection criteria using only background events, the results at fixed background acceptance levels are investigated. Taking the $O_{T_1}$ at 10 TeV as an example, the cases are considered when 1%, 5% and 10% background events are preserved. The cross sections after cuts, along with the fittings are shown in Fig. 9. For the cases of 1% with LOF, and 10% with NIF, the cross sections cannot be well fitted, and the results are not shown. For NLOF, the expected constraints at $S_{stat} = 2$ are $[-3.66 \times 10^{-3}, 1.36 \times 10^{-3}]$ (TeV$^{-4}$), $[-4.67 \times 10^{-3}, 2.28 \times 10^{-3}]$ (TeV$^{-4}$) and $[-2.25 \times 10^{-3}, 2.83 \times 10^{-3}]$ (TeV$^{-4}$), for the cases of 1%, 5% and 10%, respectively. As can be seen, the obtained coefficient constraints exhibit a spin comparable in magnitude to the case when the optimal $\Delta a_{th}$ is chosen, demonstrate the feasibility of the NLOF with information of only background events. However, the coefficient constraints obtained in this way are systematically less stringent. As a comparison, the coefficient constraints for the cases of 1%, 5% and 10% background efficiencies obtained by traditional method, LOF, IF, and NIF are shown in the right panel if Fig. 8. The NLOF is the tightest among all cases.

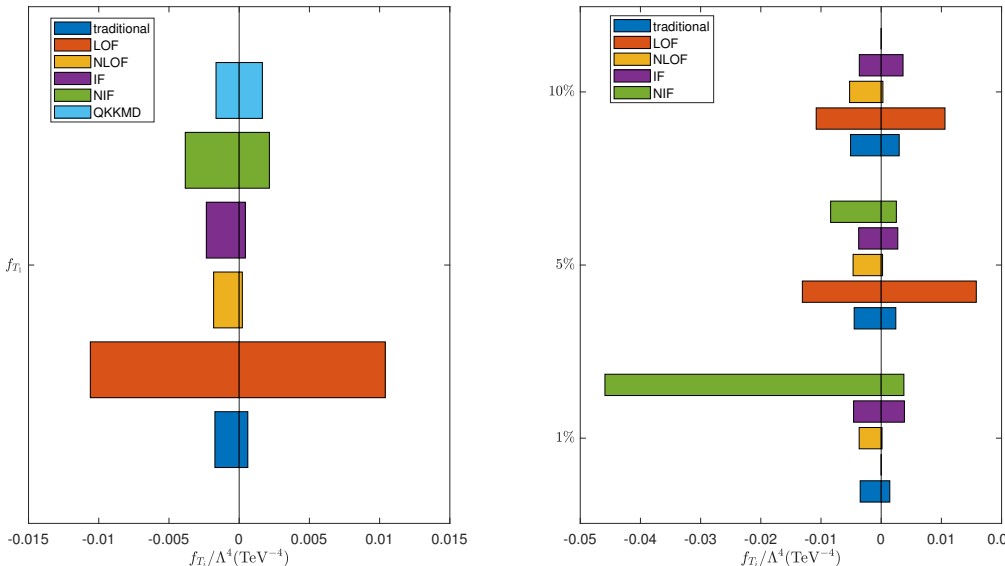

Figure 8: The left figure shows the comparison of the expected coefficient constraint when $S_{stat} = 2$ for $O_{T_1}$ at $\sqrt{s} = 10$ TeV obtained by traditional methods, LOF, NLOF, IF, NIF, and QKKMD when the event selection strategies are optimized according to signal significance. The right figure shows the comparison of the expected coefficient constraint when $S_{stat} = 2$ for $O_{T_1}$ at $\sqrt{s} = 10$ TeV obtained by traditional methods, LOF, NLOF, IF, and NIF when the background is suppressed to 1%, 5%, and 10%. For the cases of 1% with LOF, and 10% with NIF, the cross sections cannot be well fitted, and the results are not shown.

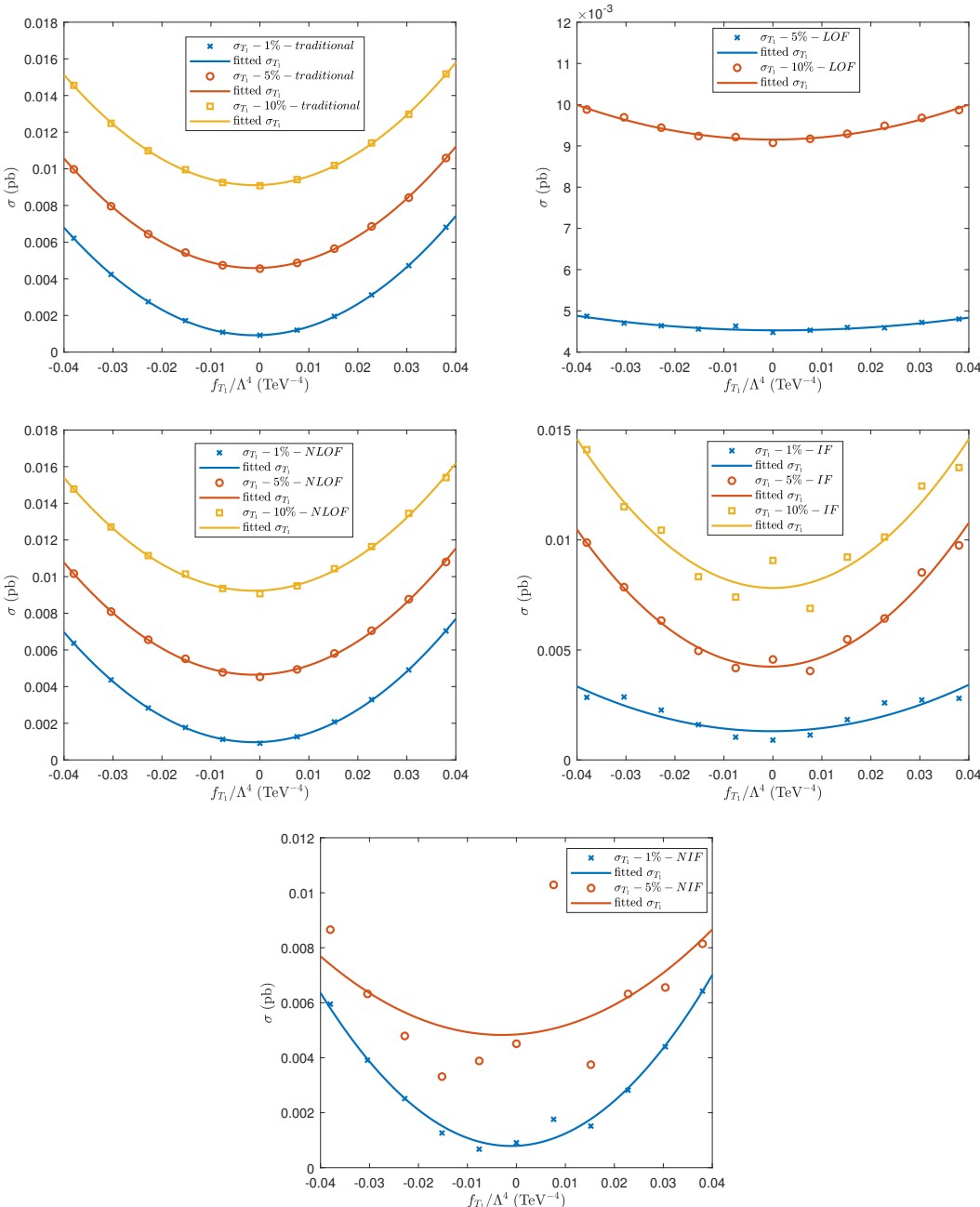

Figure 9: Taking the case of $O_{T_1}$ at $\sqrt{s} = 10$ TeV as an example, the fitted cross sections after cuts with the traditional methods (the left panel in the first row), LOF (the right panel in the first row), NLOF (the left panel in the second row), IF (the right panel in the second row), and NIF (the panel in the third row), when the background efficiency is fixed to 1%, 5%, and 10%. For the cases of 1% with LOF, and 10% with NIF, the cross sections cannot be well fitted, and the results are not shown.

# 5 Summary

In recent years, with the increasing luminosities of colliders, handling the growing amount of data has become a major challenge for future NP phenomenological research. To improve efficiency, ML algorithms have been introduced into the field of high-energy physics, including the the (N)LOF algorithms. This paper investigates how to search for NP signals using (N)LOF anomaly detection event selection strategy. Taking the process $\mu^+\mu^- \to \gamma\gamma\nu\bar{\nu}$ at muon colliders as an example, the dimension-8 operators contributing to aQGCs are studied. Expected coefficient constraints obtained using (N)LOF algorithm are presented.

The results indicate that, the VBS process $\mu^+\mu^- \to \gamma\gamma\nu\bar{\nu}$ at muon colliders is sensitive to the aQGCs. It can be concluded that, the (N)LOF can contribute to the search of signals from aQGCs. It is shown that the NLOF algorithm can improve the sensitivity of the search for aQGCs by about one order of magnitude compared to the traditional local outlier factor (LOF) algorithm. The expected coefficient constraints obtained using NLOF algorithm are shown to be tighter than KMAD, QKMAD, and a tradition counterpart.

In the case of $k \ll m$ and $d \ll m$, where $m$ is the size of dataset, $d$ is the dimension of feature space, with k-dimensional tree to calculate the k-nearest neighbors, the computational complexity to calculate the LOF anomaly scores of a dataset can be estimated as $\mathcal{O}(dm\log(m)) + \mathcal{O}(m\log m) + \mathcal{O}(mk)$. In the case of NLOF, two datasets are needed. Assuming $m_1$ is the size of test dataset, $m_2$ is the size of reference dataset (the SM background events). After calculating the anomaly scores of the two datasets, one needs to find for each point of test dataset, the nearest point in reference dataset to that point, where the computational complexity is $\mathcal{O}(d(m_1 + m_2)\log m_2)$. So, in the case of NLOF, the dominant cost is still the calculation of the anomaly scores, and the computational complexity is about $\mathcal{O}((d+1)m_1\log(m_1) + ((2d+2)m_2 + dm_1)\log(m_2))$ when $k \ll m_{1,2}$. Since the reference dataset is obtained from MC, and does not grow with the luminosities of the colliders, by assuming $m_2 \ll m_1$, the increasing of computational complexity is moderate to use NLOF instead of LOF.

As a density-based algorithm, the core computation in LOF primarily involves calculating point-to-point distances. Even when extended with nesting, as in the nested LOF (NLOF) algorithm proposed in this study, the computational backbone remains anchored in distance calculations. This grants both LOF and NLOF inherent flexibility, i.e., we can strategically define various kernel functions, precompute inter-point distances, and subsequently input them within (N)LOF frameworks. Notably, with the recent surge of quantum ML applications in NP searches, quantum computing, as a high-throughput data processing paradigm, enables ultra-efficient distance computation through quantum kernels. This naturally facilitates quantum-enhanced extensions, quantum kernel (N)LOF in the future.

**Funding information** This work was supported in part by the National Natural Science Foundation of China under Grants Nos. 11905093 and 12147214, and was supported in part by National Key R&D Program of China under Contracts No. 2022YFE0116900, and by the Natural Science Foundation of the Liaoning Scientific Committee Nos. JYTMS20231053 and LJKMZ20221431.

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
