# Peer review of "Search for anomalous quartic gauge couplings in the process $\mu^+\mu^-\to \bar{\nu}\nu\gamma\gamma$ with a nested local outlier factor"

_SciPost Physics Core_

## Round 1 · Referee Report · Anonymous (Referee 1) · 2025-8-30

Strengths

  1. The physics case is relevant: muon colliders are increasingly discussed as next-generation machines, and aQGCs at dimension-8 are a natural test ground.

  2. NLOF is a new and fresh idea in this context. The paper reasonably explains how LOF and NLOF work, which is useful for readers not familiar with these algorithms.

  3. The results are well illustrated, with fitted cross sections and clear comparisons between different methods.

  4. The focus on dimension-8 operators (disentangling quartic from triple gauge couplings) is well motivated.

Weaknesses

  1. Detector realism is too limited. The study uses a simple Delphes card with no detailed discussion of photon identification efficiencies or systematic uncertainties. This makes the quoted sensitivities likely too optimistic.

  2. Benchmarking is weak. The paper compares NLOF mainly to LOF, k-means, and a simple cut-based strategy. However, ATLAS and CMS have already explored more advanced unsupervised methods such as autoencoders, and normalizing flows. Without at least discussing these, the paper risks overstating the originality of NLOF algorithm.

  3. SMEFT validity is not fully addressed. The authors include unitarity bounds but do not discuss the EFT breakdown at 10 TeV in detail. Some quoted bounds are so tight that it is questionable whether SMEFT is reliable in that region.

  4. Hyperparameter tuning. The NLOF thresholds and neighborhood size are chosen somewhat ad hoc, with no systematic scan. This raises questions about reproducibility and robustness.

  5. How could NLOF be implemented efficiently for realistic event counts. Is it scalable to many events ?

Report

The paper investigates anomalous quartic gauge couplings (aQGCs) at a future muon collider. The novelty lies in applying the Nested Local Outlier Factor (NLOF) algorithm as an anomaly detection method, comparing its performance to LOF, k-means, quantum kernel methods, and traditional cut-based strategies. The authors find that NLOF can improve sensitivity by up to an order of magnitude, leading to stronger constraints on Wilson coefficients.

Requested changes

  1. Provide a more realistic discussion of detector effects and systematics, especially muon-collider specific challenges.

  2. Compare NLOF (at least in discussion) with SOTA anomaly detection methods already studied and used by ATLAS/CMS (autoencoders, flows, weak supervision).

  3. Add a robustness check: how do results change when varying k, thresholds, or feature choices?

  4. Discuss EFT validity at 10 TeV more carefully, including where unitarity bounds suggest SMEFT may not be trustworthy.

  5. Comment on scalability of NLOF.

Recommendation

Ask for major revision

---

## Round 1 · Referee Report · Anonymous (Referee 2) · 2025-9-1

Strengths

These has been added in the attached report.

Weaknesses

These has been added in the attached report.

Report

As indicated in the attached report, I recommend a major revision before the manuscript can be considered for acceptance in SciPost Physics.

Requested changes

Reported in the attached report.

Attachment

Recommendation

Ask for major revision

---

## Round 2 · Referee Report · Anonymous (Referee 2) · 2025-10-26

Report

I have reviewed the revised manuscript, and the authors have satisfactorily addressed all my questions and comments. I therefore recommend acceptance for publication in SciPost.

Recommendation

Publish (meets expectations and criteria for this Journal)

---

## Round 2 · Referee Report · Anonymous (Referee 1) · 2025-11-16

Report

Thank you for the revised manuscript and for providing detailed responses to the previous questions. The paper reads much more clearly now. The flow from the physics setup, to the NLOF method, to the numerical results and comparisons is well structured. The added parts: the complexity comparison between NLOF and LOF, the robustness checks, and the background-only threshold study definitely improve the work.
I have a few remaining suggestions:
1. About the “unsupervised” terminology:
I still think the wording in the Abstract/Introduction should be softened a bit. LOF itself is an unsupervised algorithm, but the overall pipeline used here (threshold optimisation, coefficient extraction, etc.) is clearly not fully unsupervised.
It would help if the authors explicitly said something like “unsupervised anomaly score, with supervised optimisation for EFT sensitivity” at least once, just to avoid giving the impression that the entire analysis is model-agnostic.
2. Connection to autoencoder-based approaches:
Since the paper already mentions using dimensionality reduction or latent-space methods, it would be good to briefly note (e.g. in the Introduction or Outlook) that NLOF could naturally be combined with the AE-based aQGC studies already in the literature (such as Ref. [21]). A short remark on how NLOF might behave in a learned latent space would make the paper more complete.
3. Detector realism and systematics:
I would encourage adding a more explicit paragraph on detector effects and systematics. In particular:
– State clearly that the results are purely statistical and do not include detector systematics or beam-induced backgrounds.
– Add a short qualitative comment (with references) on photon-ID efficiency, resolutions, and how these could affect the chosen observables.
This can be done textually — no need for new MC — but the current “we neglect these backgrounds” is too brief.
4. Minor editorial corrections:
– “a tradition counterpart” → “a traditional counterpart”
– “compare of NLOF” → “comparison of NLOF”
– “exhibit a spin comparable” → “exhibit a span comparable” (I assume “span” is meant)
Thank you very much.

Attachment

Recommendation

Ask for minor revision

---

## Round 2 · Author Response

Warnings issued while processing user-supplied markup:

  • Inconsistency: plain/Markdown and reStructuredText syntaxes are mixed. Markdown will be used.
    Add "#coerce:reST" or "#coerce:plain" as the first line of your text to force reStructuredText or no markup.
    You may also contact the helpdesk if the formatting is incorrect and you are unable to edit your text.

Dear Editor,

We are grateful to the referee for the careful review of our manuscript and for the constructive and valuable comments on ``Search for anomalous quartic gauge couplings in the process $\mu^+\mu^-\to \bar{\nu}\nu\gamma\gamma$ with a nested local outlier factor''(scipost_202507_00021v1). We appreciate the opportunity to revise our manuscript, and sincerely appreciate the helpful suggestions that reviewer has provided, which have significantly enhanced the quality of our manuscript. The following is the one-to-one response.

Responses to report 1:

  1. Provide a more realistic discussion of detector effects and systematics, especially muon-collider specific challenges.

Thank you very much for the question. We have emphasis the statement that the detector simulation is carried out using the default muon collider card in the first paragraph of section 4.1.

This is indeed a crude simulation, as the current Delphes does not incorporate the full suite of detector effects~(given that the muon colliders are not built yet). At the muon colliders, the VBS processes are associated with very energetic muons or neutrinos in the forward region with respect to the beam. One important background not included in Delphes could be the beam induced background, especially those with the final states containing soft/collinear photons/muons which can escape from the detectors and mimic the neutrinos. It is also challenging to generate events with soft/collinear final states using Monte Carlo methods, as they often lead to infrared divergences, making such backgrounds difficult to study. In this work, we neglect the contribution from these backgrounds. We have added the discussions at the beginning of section 2.

  1. Compare NLOF (at least in discussion) with SOTA anomaly detection methods already studied and used by ATLAS/CMS (autoencoders, flows, weak supervision).

Thank you for the suggestion. Isolation forest~(IF) was used to study the signals of aQGCs in VBS processes at the LHC. Nested IF~(NIF) was employed to investigate the signals of neutral triple gauge couplings~(nTGCs) in diboson processes at the CEPC. In the revision, we also utilize IF and NIF to study the $O_{T_1}$ operator in the process $\mu^+\mu^-\to \gamma\gamma\nu\bar{\nu}$ at $\sqrt{s}=10\;{\rm TeV}$. Besides, the QKMAD was used to investigate the very same process and same NP model as this work, which also provides a straightforward comparison. It can be seen from the comparison that, the expected coefficient constraint of the NLOF algorithm is the tightest among all methods. The results of IF/NIF are added in section 4.4 in the revision.

  1. Add a robustness check: how do results change when varying k, thresholds, or feature choices?

Thank you very much for the suggestion. In the article, we take $O_{T_1}$ as an example to present the dependence of expected constraints on the choice of $k$. It can be observed that larger $k$-values yield tighter coefficient constraints. Even larger $k$-values are not considered due to computational resource limitations. However, based on the comparison , it can be inferred that the sensitivity of the coefficient constraints to $k$ is moderate. We have added the comparison in section 4.3.

Another hyperparameter is the choice of $\Delta a_{th}$. As an example, the expected constraints on $O_{T_1}$ at $\sqrt{s}=10\;{\rm TeV}$ as a function of $\Delta a_{th}$ is shown. It can be seen that, the thresholds of the anomaly scores to select the events are optimized for signal significance. We have added the analysis in section 4.3.

In our work, the original feature space consists of a 12-dimensional space composed of the four-momenta of the two final-state photons and the missing four-momentum. In order to reduce the 12-dimensional feature space to 5 dimensions, we analyzed the final state and selected five observables as the feature space. Such dimensionality reduction can, in fact, be achieved in a process-agnostic manner using ML algorithms, automatically and without the need to analyze the underlying physics. The discussions are added at the end of section 4.1. The primary reason for performing dimensionality reduction is actually due to limitations in computational power. As $k$ increases, the computational resources required also increase. Meanwhile, a larger $k$ generally leads to better performance in LOF/NLOF. To enable the computation of larger $k$ values, we opted for a lower-dimensional feature space.

  1. Discuss EFT validity at 10 TeV more carefully, including where unitarity bounds suggest SMEFT may not be trustworthy.

Thank you very much to point out this issue. The validity of the EFT is indeed of significant importance in the study of SMEFT. In the revisions, we have supplemented the details on partial-wave unitarity bounds at the end of section 2.

There was a typo in our Table 5, where we mistakenly placed $10^-4$ in the $\mathcal{S}_{stat}$ column (it should actually be in the coefficient constraint column). We have corrected this error in the revised version. All the expected coefficient constraints obtained in this work remain within the partial-wave unitarity bounds.

  1. Comment on scalability of NLOF.

This suggestion you put forward is highly valuable. In the case of $k\ll m$ and $d\ll m$, where $m$ is the size of dataset, $d$ is the dimension of feature space, with k-dimensional tree to calculate the k-nearest neighbors, the computational complexity to calculate the LOF anomaly scores of a dataset can be estimated as $\mathcal{O}(dm\log (m))+ \mathcal{O}(m\log m) + \mathcal{O}(mk)$. In the case of NLOF, two datasets are needed. Assuming $m_1$ is the size of test dataset, $m_2$ is the size of reference dataset~(the SM background events). After calculating the anomaly scores of the two datasets, one needs to find for each point of test dataset, the nearest point in reference dataset to that point, where the computational complexity is $\mathcal{O}(d(m_1 + m_2) \log m_2)$. So, in the case of NLOF, the dominant cost is still the calculation of the anomaly scores, and the computational complexity is about $\mathcal{O}((d+1)m_1\log (m_1)+\left((2d+2)m_2+dm_1\right)\log (m_2))$ when $k\ll m_{1,2}$. Since the reference dataset is obtained from MC, and does not grow with the luminosities of the colliders, by assuming $m_2\ll m_1$, the increasing of computational complexity is moderate to use NLOF instead of LOF. We have added the discussions in the third paragraph of section 5.

Responses to report 2:

  1. The method is presented as fully unsupervised, yet in Sect. 4 the thresholds $a_{th}$ and $\Delta a_{th}$ and the choice of $k$ are optimized using signal significance. This introduces a degree of supervision, effectively tailoring the method to the specific signal model under study (dimension-8 aQGCs). True unsupervised anomaly detection should define thresholds from background data alone. I strongly recommend that the authors clarify this point, and ideally include results where thresholds are set purely from background distributions and signal is used only for evaluation.

Thank you very much for the question, this is an important point. We have added in the fifth paragraph of the introduction to clarify this point. It is worth clarifying that if the goal were solely to identify anomalous events, one would not need to know the NP model. However, if no trace of NP were found, the purpose of NP phenomenology becomes constraining the parameters of NP. To achieve this, introducing an NP model becomes necessary. In our phenomenological study, not only the aQGCs are introduced, but also an event selection strategy designed to maximize the signal is adopted, which incorporates supervised learning. To investigate the impact of developing event selection criteria using only background events, the scenarios are also considered where the number of remaining background events is $1\%$, $5\%$, and $10\%$ (as suggested in your question 4).

  1. NLOF is compared against LOF, k-means anomaly detection (KMAD/QKMAD), and cut-based selections. These are not the most relevant state-of-the-art baselines. Since NLOF is a density-based method, more appropriate comparisons would include DBSCAN, OPTICS, Isolation Forest, kNN-based distances, etc ... Even a limited comparison on a subset of these would provide a much more convincing demonstration that NLOF offers genuine improvements.

Thank you for the suggestion. Isolation forest~(IF) was used to study the signals of aQGCs in VBS processes at the LHC. Nested IF~(NIF) was employed to investigate the signals of neutral triple gauge couplings~(nTGCs) in diboson processes at the CEPC. In the revision, we also utilize IF and NIF to study the $O_{T_1}$ operator in the process $\mu^+\mu^-\to \gamma\gamma\nu\bar{\nu}$ at $\sqrt{s}=10\;{\rm TeV}$. The same dataset is used, and the forest consists of $100$ trees. For both IF and NIF, the criteria are optimized for the signal significance. It can be seen from the comparison that, the expected coefficient constraint of the NLOF algorithm is the tightest among all methods. The results are added in section 4.4 in the revision.

  1. The current results focus only on cross sections after optimized cuts and the resulting EFT coefficient bounds. It would be very valuable to also show ROC curves and AUC values for LOF, NLOF, k-means, and cut-based selections. These are threshold-independent and directly illustrate the separation power of the anomaly scores. This would also mitigate the concern about ``cheating'' via signal-based cut optimization.

Thank you very much for the suggestion.

Indeed, ROC and AUC are important metrics for comparing classification or anomaly detection algorithms. However, LOF/NLOF does not learn from background data to select the signal events; it requires learning from a dataset containing both signal and background events, where the contribution from the interference term cannot be traced. While one could ignore the interference term and mix pure SM and pure NP events according to the ratio of their cross sections to study ROC and AUC, the results from the coefficient constraints indicate that the interference term plays a significant role. Therefore, we maintain that the additional effort required for such an approach would not accurately reflect the discriminative power of LOF/NLOF. Furthermore, in our comparisons, all methods, whether traditional approaches, KMAD/QKMAD, or the newly added IF/NIF, underwent signal-based cut optimization using Monte Carlo data. This ensures a fair comparison, free from the "cheating" issue mentioned. Consequently, we have not implemented this suggested modification in our revision.

  1. In Sec. 4 the anomaly score thresholds are tuned to maximize significance. This mixes background rejection and signal efficiency. To make the comparison more transparent, I recommend showing significance values at fixed background acceptance levels (e.g. 1\%, 5\%, 10\%). This would highlight the gain in signal efficiency provided by NLOF relative to LOF or k-means, independent of cut tuning.

Thank you very much for this valuable suggestion. To make the comparison more transparent, and to study the impact of developing event selection criteria using only background events, the results at fixed background acceptance levels are investigated. Taking the $O_{T_1}$ at $10\;{\rm TeV}$ as an example, the cases are considered when $1\%$, $5\%$ and $10\%$ background events are preserved. For NLOF, the expected constraints at $S_{stat}=2$ are $\left [ -3.66\times 10^{-3},1.36\times 10^{-3} \right ]\;(\rm TeV^{-4})$, $\left [ -4.67\times 10^{-3},2.28\times 10^{-3} \right ]\;(\rm TeV^{-4})$ and $\left [ -2.25\times 10^{-3},2.83 \times 10^{-3}\right ]\;(\rm TeV^{-4})$, for the cases of $1\%$, $5\%$ and $10\%$, respectively. As can be seen, the obtained coefficient constraints exhibit a spin comparable in magnitude to the case when the optimal $\Delta a_{th}$ is chosen, demonstrate the feasibility of the NLOF with information of only background events. However, the coefficient constraints obtained in this way are systematically less stringent. As a comparison, the coefficient constraints for the cases of $1\%$, $5\%$ and $10\%$ background efficiencies obtained by traditional method, LOF, IF, and NIF are shown, and the NLOF is the tightest among all cases. We have added the analysis at the end of section 4.4.

  1. A natural and widely used extension is to train an autoencoder on SM data and apply anomaly detection (eg. LOF, NLOF) in the learned latent space. This often improves robustness and separation power. Even a short discussion of this possibility would broaden the impact of the work.

This suggestion you put forward is highly valuable. Autoencoders can be utilized for anomaly detection, with one approach involving the application of other classification or anomaly detection algorithms, such as LOF/NLOF, in the latent space. Employing LOF/NLOF in the latent space is equivalent to performing dimensionality reduction before the using of LOF/NLOF. In our work, the original feature space consists of a 12-dimensional space composed of the four-momenta of the two final-state photons and the missing four-momentum. In order to reduce the 12-dimensional feature space to 5 dimensions, we analyzed the final state and selected five observables as the feature space. Such dimensionality reduction can, in fact, be achieved in a process-agnostic manner using ML algorithms, automatically and without the need to analyze the underlying physics. For instance, autoencoders or principle component analysis~(PCA) can be employed for data dimensionality reduction. For complex processes with a large number of final-state particles, where manual analysis for dimensionality reduction becomes particularly intricate, this approach is especially meaningful. We have added a discussion on this in the revised version at the end of section 4.1.

  1. Figures 2-4 show cross sections and coefficient bounds after optimized cuts. These effectively fold in the anomaly detection performance, but they do not directly show how well LOF/NLOF separate signal from background. Adding anomaly score distributions and ROC curves alongside these figures would make the connection clearer.

This suggestion is also highly valuable. The distribution of the anomaly scores is frequently presented in machine learning studies of new physics signals, as it not only illustrates the separation between signal and background more clearly but also helps determine the optimal threshold position. However, this aspect closely aligns with the concerns already raised in your Question 3. To do this, we need a dataset mixing pure SM and pure NP events according to the ratio of their cross sections. However, such a dataset does not reflect the truth of the anomaly scores, which needs the learning of the whole dataset to be determined, because the contribution of the interference term is not negligible. Consequently, we have not implemented this suggested modification in our revision.

  1. Finally it would be helpful to state the computational cost of NLOF relative to LOF.

Thank you very much for the question. In the case of $k\ll m$ and $d\ll m$, where $m$ is the size of dataset, $d$ is the dimension of feature space, with k-dimensional tree to calculate the k-nearest neighbors, the computational complexity to calculate the LOF anomaly scores of a dataset can be estimated as $\mathcal{O}(dm\log (m))+ \mathcal{O}(m\log m) + \mathcal{O}(mk)$. In the case of NLOF, two datasets are needed. Assuming $m_1$ is the size of test dataset, $m_2$ is the size of reference dataset~(the SM background events). After calculating the anomaly scores of the two datasets, one needs to find for each point of test dataset, the nearest point in reference dataset to that point, where the computational complexity is $\mathcal{O}(d(m_1 + m_2) \log m_2)$. So, in the case of NLOF, the dominant cost is still the calculation of the anomaly scores, and the computational complexity is about $\mathcal{O}((d+1)m_1\log (m_1)+\left((2d+2)m_2+dm_1\right)\log (m_2))$ when $k\ll m_{1,2}$. Since the reference dataset is obtained from MC, and does not grow with the luminosities of the colliders, by assuming $m_2\ll m_1$, the increasing of computational complexity is moderate to use NLOF instead of LOF. We have added the discussions in the third paragraph of section 5.

Our primary revisions are marked in red. We hope that our responses and the revised manuscript address the concerns.

With Best Regards.

Yours sincerely.

Ke-Xin Chen, Yu-Chen Guo and Ji-Chong Yang

Department of Physics, Liaoning Normal University, Dalian 116029, China

---

## Round 2 · List of Changes

We have emphasis the statement that the detector simulation is carried out using the default muon collider card in the first paragraph of section 4.1.
We have added the discussion on muon-collider specific challenges on detector effects at the beginning of section 2.
The results of IF/NIF are added in section 4.4 for a comparison in the revision.
We have added the comparison for different $k$ in section 4.3.
We have added the analysis on the choice of $\Delta a_{th}$ section 4.3.
The discussions on dimension reduction/latent space of autoencoder are added at the end of section 4.1.
We have supplemented the details on partial-wave unitarity bounds at the end of section 2.
We have corrected this errors in Table 5 in the revised version.
We have added the discussions on the computational complexity in the third paragraph of section 5.
We have added in the fifth paragraph of the introduction to clarify the issue on supervised learning.
We have added the analysis at fixed background acceptance levels (1\%, 5\%, 10\%) the at the end of section 4.4.

Our primary revisions are marked in red.

---

## Round 3 · Author Response

Dear Editor,

We are grateful to the referee for the careful review of our manuscript and for the constructive and valuable comments on ``Search for anomalous quartic gauge couplings in the process $\mu^+\mu^-\to \bar{\nu}\nu\gamma\gamma$ with a nested local outlier factor''(scipost_202507_00021v1). We appreciate the opportunity to revise our manuscript, and sincerely appreciate the helpful suggestions that reviewer has provided, which have significantly enhanced the quality of our manuscript. The following is the one-to-one response.

  1. About the unsupervised'' terminology: I still think the wording in the Abstract/Introduction should be softened a bit. LOF itself is an unsupervised algorithm, but the overall pipeline used here (threshold optimisation, coefficient extraction, etc.) is clearly not fully unsupervised. It would help if the authors explicitly said something likeunsupervised anomaly score, with supervised optimisation for EFT sensitivity'' at least once, just to avoid giving the impression that the entire analysis is model-agnostic.

Thank you very much for the suggestion. We have emphasis the statement of the ``supervised optimisation'' in abstract and in the second paragraph of introduction.

  1. Connection to autoencoder-based approaches: Since the paper already mentions using dimensionality reduction or latent-space methods, it would be good to briefly note (e.g. in the Introduction or Outlook) that NLOF could naturally be combined with the AE-based aQGC studies already in the literature (such as Ref. [21]). A short remark on how NLOF might behave in a learned latent space would make the paper more complete.

Thank you for the suggestion. Since the main computational cost of the LOF/NLOF algorithm lies in calculating distances, it can be anticipated that applying dimensionality reduction and implementing LOF/NLOF in the latent space will significantly reduce computational complexity and improve the algorithm's operational efficiency. We have added the statement in the third paragraph in section 4.1.

  1. Detector realism and systematics: I would encourage adding a more explicit paragraph on detector effects and systematics. In particular:\
  2. State clearly that the results are purely statistical and do not include detector systematics or beam-induced backgrounds.
  3. Add a short qualitative comment (with references) on photon-ID efficiency, resolutions, and how these could affect the chosen observables. This can be done textually - no need for new MC - but the current ``we neglect these backgrounds'' is too brief.

Thank you very much for the suggestion.

Since the muon collider is a future collider, at this stage the effect of the detector can only be estimated using simulation. In this work, we choose the default muon collider card in Delphes. That means, the results are purely statistical and do not include detector systematics or beam-induced backgrounds. Denoting $E_{\gamma}$ and $\eta _{\gamma}$ as the energy and pseudo-rapidity of a photon, the photon efficiency is $94\%$ when $E_{\gamma}\geq 2\;{\rm GeV}$ and $|\eta_{\gamma}|<0.7$, and $90\%$ when $E_{\gamma}\geq 2\;{\rm GeV}$ and $2.5\geq |\eta_{\gamma}|\geq 0.7$, and otherwise zero. The photon is considered as isolated when $\sum p_T/p_T^{\gamma} < 0.2$ where the sum runs over other particles in the vicinity of the photon with $\Delta R<0.1$ and with $p_T>0.5\;{\rm GeV}$, where $p_T$ is the transverse momentum, and $\Delta R=\sqrt{\Delta \eta^2+\Delta \phi^2}$ where $\Delta \eta$ and $\Delta \phi$ are differences of pseudo-rapidities and azimuth angles of two particles. The photon energy resolution is parameterized as $\sigma(E)/E = \sqrt{(1\%)^2 + (st^2/\sqrt{E}) }$, with $st$ denoting the stochastic term varying across three pseudo-rapidity regions: $15.6\%$ for $|\eta| \leq 0.78$, $17.5\%$ for $0.78 < |\eta| \leq 0.83$, and $15.1\%$ for $0.83 < |\eta| \leq 2.5$. We expect that, as machine learning algorithms, LOF/NLOF are insensitive to these parameters.

We have added the statement as the second paragraph in section 4.1.

  1. Minor editorial corrections:
  2. a tradition counterpart'' $\to$a traditional counterpart''
  3. compare of NLOF'' $\to$comparison of NLOF''
  4. exhibit a spin comparable'' $\to$exhibit a span comparable'' (I assume ``span'' is meant)

Thank you very much for the careful reading of our manuscript. We have revised those minor mistakes in the revision.

Our primary revisions are marked in red. We hope that our responses and the revised manuscript address the concerns.

With Best Regards.

Yours sincerely.

Ke-Xin Chen, Yu-Chen Guo and Ji-Chong Yang Department of Physics, Liaoning Normal University, Dalian 116029, China

---

## Round 3 · List of Changes

1. We have softened the statement on ``unsupervised'' in both abstract and introduction.

  2. We have added the note to encourage a latent space LOF/NLOF in the third paragraph in section 4.1 .

  3. We have added the statement on detector simulation as the second paragraph in section 4.1.

  4. Minor revisions are made for the linguistic errors.

---

## Editorial Decision

accepted_in_target_journal